# Maternal supplementation with *Bifidobacterium breve* M-16V prevents their offspring from allergic airway inflammation accelerated by the prenatal exposure to an air pollutant aerosol

Chiyoko Terada-Ikeda[1☯], Masahiro Kitabatake[1☯], Akari Hiraku[2], Kumiko Kato[3], Satsuki Yasui[1], Natsuko Imakita[1], Noriko Ouji-Sageshima[1], Noriyuki Iwabuchi[2], Kaoru Hamada[4], Toshihiro Ito[1] *

1 Department of Immunology, Nara Medical University, Kashihara, Nara, Japan, 2 R&D Division, Food Ingredients and Technology Institute, Morinaga Milk Industry Co., Ltd., Zama, Japan, 3 R&D Division, Next Generation Science Institute, Morinaga Milk Industry Co., Ltd., Zama, Japan, 4 Department of Clinical and Investigative Medicine, Faculty of Nursing, Nara Medical University, Kashihara, Nara, Japan

☯ These authors contributed equally to this work.
* toshi-ito@naramed-u.ac.jp

## Abstract

*Bifidobacterium breve* M-16V is a probiotic bacterial strain with efficacy in infants achieved by suppressing T-helper type (Th) 2 immune responses and modulating the systemic Th1/Th2 balance. Exposure to air pollution during pregnancy increases asthma susceptibility in offspring. The aim of this study was to investigate the effects of the maternal intake of *B. breve* M-16V on susceptibility to asthma accelerated by prenatal exposure to air pollution. The intake of *B. breve* M-16V in residual oil fly ash (ROFA)-exposed pregnant mice resulted in fewer eosinophils in the bronchoalveolar lavage fluid of neonatal mice and reduced allergic lung inflammation. The expressions of Th2 cytokines including IL-5 and IL-13 were decreased in neonatal mice from ROFA-exposed mothers fed *B. breve* M-16V. The analysis of fecal microbiota from neonatal mice revealed that the intake of *B. breve* M-16V by mothers changed the composition of fecal microbiota in neonatal mice, which resulted in a decreased population of *Firmicutes*. Moreover, several bacterial strains of fecal microbiota from neonatal mice had a strong correlation with Th2 cytokines and histological score. These results suggest that the maternal intake of M-16V might have beneficial effects in neonates by preventing and/or alleviating allergic reactions accelerated by prenatal exposure to air pollution.

## Introduction

The increasing incidence of asthma in early life suggests that prenatal environmental exposure may influence susceptibility to allergic airway diseases [1–3]. Indeed, recent studies suggested

**Data Availability Statement:** DNA sequences corresponding to the 16S rRNA gene data have been deposited in DDBJ under accession number DRA010221.

**Funding:** This study was supported by research grants from Morinaga Milk Industry Co., Ltd. The funder provided support in the form of salaries for A. Hiraku, K. Kato and N. Iwabuchi, but did not have any additional role in the study design, data collection and analysis, decision to publish, or preparation of the manuscript. The specific roles of these authors are articulated in the 'author contributions' section.

**Competing interests:** All authors have read the manuscript and have approved this submission. The authors have the journal's policy and have the following conflicts: A. Hiraku, K. Kato and N. Iwabuchi are employees of Morinaga Milk Industry Co., Ltd. The Morinaga Milk Industry Co., Ltd. is the commercial company that sells dairy products like milk, yoghurt, infant formula and so on. The Morinaga Milk Industry Co., Ltd. has filed patent applications on aspects of this work. A. Hiraku, N. Iwabuchi, and Dr. Ito are named as inventors on patent application number WO2019/163257 (patent name: COMPOSITION FOR PREVENTING OR RELIEVING RESPIRATORY DISEASE). These do not alter our adherence to all the PLOS ONE policies on sharing data and materials.

that prenatal exposure to air pollution enhanced the risk of developing asthma in offspring [4–6]. Epidemiological and experimental studies reported that diesel exhaust particles (DEP), a major component of particulate air pollution in many urban areas, stimulated sensitization to allergens [7, 8]. In addition, we and other groups found that a surrogate for ambient air pollution, residual oil fly ash (ROFA) from an oil combustion power plant, contributed to increased asthma susceptibility in offspring [9–11].

The pathogenesis of asthma is characterized by a large influx of eosinophils and CD4[+] T-helper (Th) cells, especially Th2 cells, around lung airways [12]. Th2 cells synthesize high levels of IL-4, IL-5, and IL-13, which induce immunoglobulin (Ig) E synthesis and eosinophilic inflammation [13].

Recent studies on the properties and functionality of living micro-organisms in food have suggested that probiotics play an important role in immunological development, as well as digestive and respiratory functions [14, 15]. Numerous animal and human studies reported that the intake of probiotics such as lactobacilli and bifidobacteria inhibited allergic responses and allergic sensitization by inhibiting Th2 responses [16–18]. There are many species within the genus *Bifidobacterium*, among which *Bifidobacterium breve* is one of the most abundant in human neonates [19].

*B. breve* M-16V is a probiotic strain isolated from the fecal sample of a human baby [20]. M-16V has been used for low-birth-weight infants in Australia and Japan [21–23]. It was previously shown that the administration of M-16V to infants with food allergy and atopic dermatitis significantly improved their allergic symptoms [15, 24, 25]. M-16V administration was associated with a significant increase in the proportion of *Bifidobacteria* and a decrease in the proportion of aerobic bacteria in the fecal microbiota [15, 24]. In addition, Inoue *et al.* demonstrated that M-16V modulated the Th1/Th2 balance in OVA-immunized mice, which are commonly used as a Th2-induced bronchial asthma model [16]. However, whether M-16V affects asthma susceptibility in offspring induced by air pollution is unknown, and a detailed correlation between fecal microbiota and Th2 responses in lungs requires further study. We previously established a mouse model to investigate the potential effects of air pollution exposure during pregnancy on the susceptibility of offspring to asthma [11].

In the present study, we evaluated the effect of M-16V on the pathogenesis of allergic asthma in our experimental asthma model with prenatal exposure to air pollutants using ROFA, and analyzed bacterial organisms in the fecal microbiota that correlated with asthma responses.

## Materials and methods

### M-16V preparation

*B. breve* M-16V powder was supplied by the Morinaga Milk Industry Co., Ltd. (Kanagawa, Japan). Pure cells were freeze-dried with dextrin to achieve the desired concentration.

### Animal studies

**Animals.** Pregnant BALB/c mice obtained from Japan SLC (Hamamatsu, Japan) were maintained at 25°C with a 12-h light/dark cycle in an animal facility of the Department of Animal Resources at Nara Medical University. Mice in the control group were fed an AIN-93G-modified basal diet (Oriental Bio Service Inc., Kyoto, Japan) and mice in the M-16V group were fed a basal diet supplemented with M-16V. The M-16V group was fed approximately 0.5 g of the AIN-93G diet with the addition of 1 million viable cells of M-16V. The control group was fed 0.5 g of the AIN-93G diet with the addition of dextrin. Each group was fed separately from their usual diet. Mothers were fed during the examination period with AIN-93G

(Control) or M-16V once a day. The diet of the mothers was continued throughout the nursing time until the analysis of the offspring. Each mother was housed with her offspring in a single cage and all neonates (both male and female) were breastfed throughout the study (Fig 1). The Animal Care and Use Committee at Nara Medical University approved all animal experiments conducted in this study (approval No.11367), and all methods were carried out based on the Policy on the Care and Use of Laboratory Animals, Nara Medical University. These experiments were carried out from 2015 to 2018.

**Study protocol.** Fig 1 summarizes the experimental protocols used in this study. ROFA obtained from the precipitator unit of a local power plant was kindly provided by Dr. Lester Kobzik (Harvard School of Public Health, Boston) [11]. ROFA was suspended in phosphate buffered saline (PBS) to 100 mg/ml and sonicated for 10 min. The ROFA suspension was incubated at 37˚C with rotation for 4 h and then centrifuged at 3000 ×$g$ for 10 min. The supernatant (leachate) was removed and diluted to 50 mg/ml in PBS. As described previously [11], pregnant BALB/c mice were exposed to nebulized PBS or ROFA leachate for 30 min at days 14, 16, and 18 of pregnancy. After birth, neonatal mice were intraperitoneally sensitized to 5 μg of ovalbumin (OVA; Grade III) (Sigma Chemical, St. Louis, MO) and 0.5 mg of alum (Nacalai Tesque Inc., Kyoto, Japan) diluted in 50 μl PBS. On days 26–28 of life, neonatal mice were exposed to aerosolized OVA (1%, 10 min/day, for 3 consecutive days) or PBS.

**Pathologic analysis.** At 48 h after the final challenge, neonatal mice were euthanized by blood collection from the left ventricle of the heart following pentobarbital (Nacalai Tesque Inc.) anesthesia. The trachea was cannulated and 0.5 ml of sterile PBS was instilled to harvest the bronchoalveolar lavage fluid (BALF), which was centrifuged at 800 ×$g$ for 5 min, and resuspended in 0.5 ml of PBS. Differential cell counts were performed on cytocentrifuge slides prepared by centrifugation of samples at 800 rpm for 5 min using Cytospin 3 (Thermo Fisher Scientific Inc., Waltham, MA). The slides were fixed in 95% methanol and stained with Diff-Quik (Sysmex, Kobe, Japan), modified Wright-Giemsa stain, and then 200 cells were counted for each sample by microscopy. Macrophages, lymphocytes, neutrophils, and eosinophils were enumerated. BALF supernatants were analyzed for the measurement of cytokines (IL-4, IL-5, IL-13) using mouse ELISA kits according to the manufacturer's protocol (Thermo Fisher Scientific Inc.). After lavage, the left lobe of the lung was removed, inflated, and fixed with 4% paraformaldehyde for histological assessment. After paraffin embedding, sections for microscopy were stained with Periodic acid-Schiff (PAS) to visualize mucus production. Sections were scored based on the mucus production of the bronchi as follows: 1 –minimal (<25%), 2 –slight (<25%–50%), 3 –moderate (<50%–75%), 4 –severe (>75%), as previously described [11, 26]. The mean score from at least 10 bronchi per section was assessed as the mucin score.

## Quantitative real-time PCR (qPCR)

The right lobe of the lung was immersed in 0.2 ml RNAlater® Stabilization Solution (Thermo Fisher Scientific Inc.) overnight at 4˚C and stored at −80˚C until RNA extraction. Total RNA was isolated using NucleoSpin® RNA (MACHEREY-NAGEL GmbH & Co. KG, Düren, Germany) and stored at −80˚C. Total RNA was extracted and 1 μg of total RNA was reverse-transcribed to cDNA according to the procedure previously described [27]. qPCR was performed with TaqMan gene expression assays using a Step One™ qPCR system (Thermo Fisher Scientific Inc.). TaqMan gene expression assays for *Gapdh* (Mm99999915), *Il4* (Mm00445259), *Il5* (Mm00439646), *Il13* (Mm00434204), and *Muc5ac* (Mm0126718) were purchased from Thermo Fisher Scientific Inc. mRNA expression was analyzed by the *ΔΔ*Ct method and normalized to GAPDH expression as previously described [27].

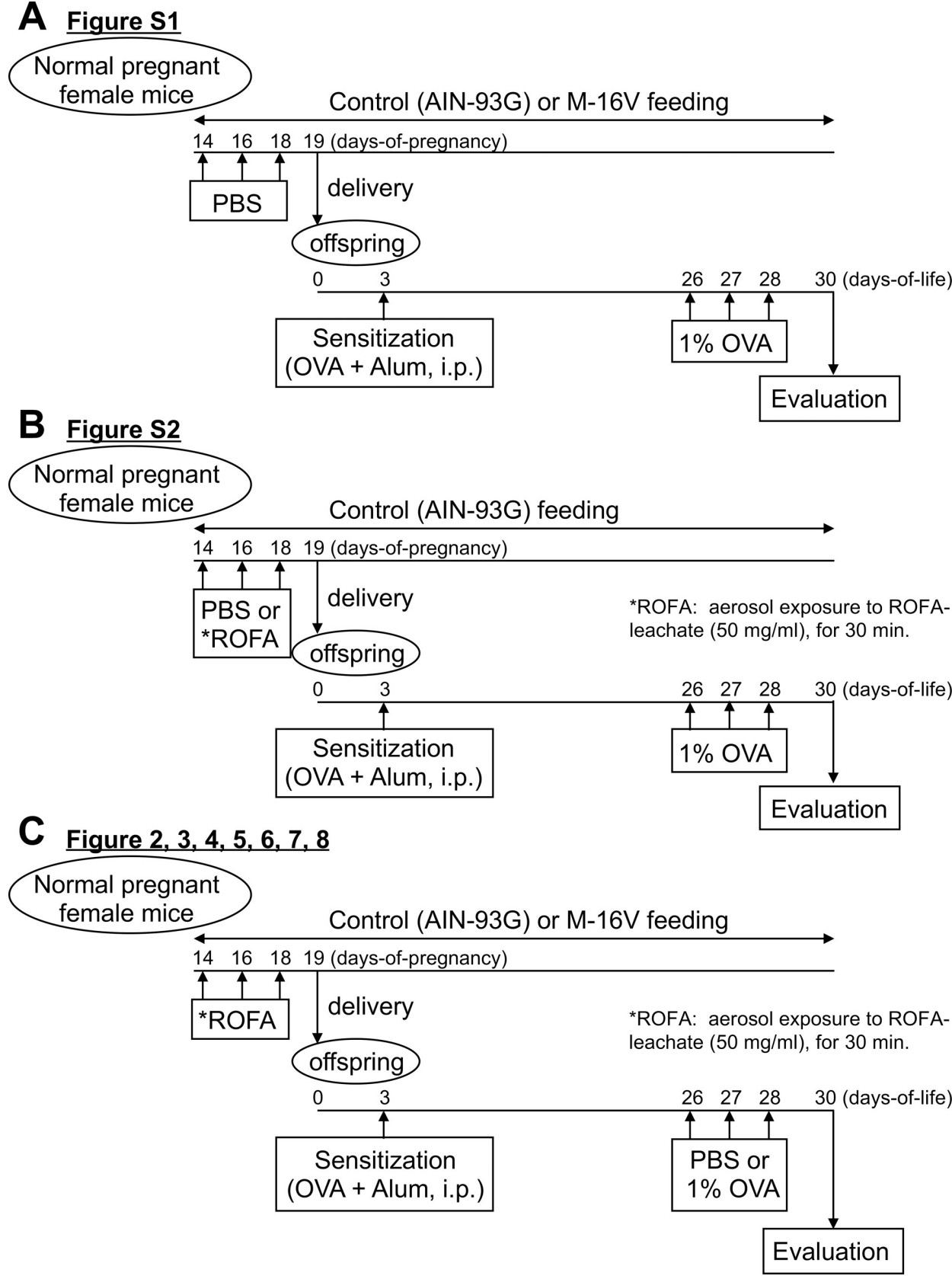

**Fig 1. Schematic representation of the experimental protocol.** (**A**) Analysis of OVA allergic neonates from mothers exposed to PBS (no ROFA) on a control (AIN-93G) or M-16V diet. On days 5, 3, and 1 before delivery, normal pregnant BALB/c female mice were exposed to aerosolized PBS. The newborns received a single intraperitoneal injection of OVA (5 µg) and alum (0.5 mg) at day 3, followed by exposure to aerosolized 1% OVA on days 26–28 of life. (**B**) Analysis of OVA allergic neonates from PBS- or ROFA-exposed mothers on a control (AIN-93G) diet. On days 5, 3, and 1 before delivery, normal pregnant BALB/c female mice were exposed to aerosolized ROFA leachate (50 mg/ml) or PBS. The newborns received a single intraperitoneal injection of OVA (5 µg) and alum (0.5 mg) at day 3, followed by exposure to aerosolized 1% OVA on days 26–28 of life. (**C**) Analysis of OVA allergic neonates from mothers exposed to ROFA on a control (AIN-93G) or M-16V diet. On days 5, 3, and 1 before delivery, normal pregnant BALB/c female mice were exposed to aerosolized ROFA leachate (50 mg/ml). The newborns received a single intraperitoneal injection of OVA (5 µg) and alum (0.5 mg) at day 3, followed by exposure to aerosolized 1% OVA or PBS on days 26–28 of life. All Analyses in this study were performed 48 h after the final aerosol exposure.

## Microbiota profiling

**DNA extraction from fecal samples.** Fecal samples were collected at 48 h after final challenge and stored at −80°C until analysis. DNA was extracted using the bead-beating method as previously described [28] with some modifications. Fecal samples (20 mg) were suspended in 450 µl extraction buffer (100 mM Tris/HCl, 40 mM EDTA, pH 9.0) and 50 µl 10% SDS. Glass beads (300 mg, 0.1 mm diameter) and 500 µl buffer-saturated phenol were added to the suspension and the mixture was vortexed vigorously for 180 s using a Multi-Beads Shocker (Yasui Kikai Co., Osaka, Japan) at a speed of 2,700 rpm. After centrifugation at 14,000 ×$g$ for 5 min, 400 µl of the supernatant was extracted with phenol-chloroform and 250 µl of the supernatant was precipitated with isopropanol. Purified DNA was suspended in 2,000 µl of Tris-EDTA buffer (pH 8.0).

**16S rRNA gene sequencing.** Subsequent DNA extraction and sequencing by Illumina Miseq (Illumina Inc., San Diego, CA) were performed as described previously [29, 30]. Briefly, the V3-V4 region of the bacterial 16S rRNA gene was amplified by PCR using the Takara Ex Taq HS Kit (Takara Bio, Shiga, Japan) and the primer sets Tru357F (5′–CGCTCTTCCGATC TCTGTACGGRAGGCAGCAG–3′) and Tru806R (5′–CGCTCTTCCGATCTGACG–GACTACHV GGGTWTCTAAT–3′) with the following protocol: preheating at 94°C for 3 min, 30 cycles of denaturation at 94°C for 30 s, annealing at 50°C for 30 s, extension at 72°C for 30 s, and terminal extension at 72°C for 5 min. Then, 1 µl sample of the PCR product was amplified using the following barcoded primers adapted for Illumina MiSeq: Fwd 5′–AATGATACGGCGACCAC CGAGATCTACACXXXXXXXXXACACTCTTTCCCTACACGACGCTCTTCCGATCTCTG–3′ and Rev 5′–CAAGCAGAAGACGGCATACGAGATXXXXXXXXXGTGACTGGAGTTCAGACGTGTGCT CTTCCGATCTGAC–3′, where X represents a barcode base. Amplification was performed according to the protocol described above except only eight cycles were performed. The products were purified using a QIAquick PCR Purification Kit (Qiagen, Valencia, CA) according to the manufacturer's protocols. The purified products were quantified by Quant-iT Pico-Green dsDNA Assay Kit (Thermo Fisher Scientific). Equal amounts of amplicons were pooled and purified with the GeneRead Size Selection Kit (Qiagen) according to the manufacturer's protocol. The pooled libraries were sequenced using an Illumina MiSeq instrument and the MiSeq v3 Reagent Kit (Illumina Inc.).

**Sequencing data analysis.** Briefly, phiX reads were removed from the raw Illumina paired-end reads and the sequences were analyzed using the QIIME software package version 1.8.0 (http://qiime.org/) [31, 32]. After removing potential chimeric sequences using UCHIME, assignment to operational taxonomic units (OTUs) using open-reference OTU picking [33] with a 97% threshold of pairwise identity was conducted. Subsequently, the taxonomical classification was performed using the Greengenes reference database (http://greengenes.secondgenome.com/downloads/database/13_5) [34]. UniFrac distances were calculated using QIIME version 1.8.0 software [31]. DNA sequences corresponding to the 16S rRNA gene data have been deposited in the DDBJ under accession number DRA010221.

## Statistical analysis

Statistical significances in the gene expression, mucin score, and BAL analysis data were evaluated by analysis of variance. P values < 0.05 were considered to indicate statistically significant differences. Statistical analyses except for the analyses of gut microbiota were performed using GraphPad Prism 4.0 (GraphPad Software, San Diego, CA). Data are presented as the mean ± SEM and are representative of at least two independent experiments. The analyses of gut microbiota without the calculation of a false discovery rate (FDR) were performed using SPSS version 23.0 statistical software (IBM Corp., Armonk, NY). Intergroup differences were analyzed using the Mann-Whitney *U*-test and correlation analysis between gut bacteria and Th2 responses in the lungs were performed by Spearman's correlation coefficient. Results were adjusted by FDR using the Benjamini and Hochberg method in R software version 3.6.0 (R Foundation for Statistical Computing, http://www.R-project.org/).

## Results

### ROFA during pregnancy accelerates allergic airway inflammation

We first compared the response of neonatal mice in the control (AIN-93G) and M-16V diet groups to OVA without ROFA exposure (Fig 1A). To examine the response in lungs, a histological assessment of lung tissue from mice euthanized at 48 h after the last challenge was conducted. The mice showed robust pathologic changes caused by allergic inflammation including an accumulation of eosinophils and mononuclear cells around the airways and vessels, and goblet cell hyperplasia. As shown in S1 Fig, there were no significant differences in mucin score or allergic Th2 gene expressions including IL-4, IL-5, IL-13, and Muc5ac between control (AIN-93G) and M-16V diet mice not exposed to ROFA. We previously reported offspring from mothers exposed to ROFA during pregnancy developed robust pathologic changes related to airway inflammation including mucin score [11]; therefore, we next assessed whether ROFA during pregnancy aggravated the response of neonatal mice to OVA in our model (Fig 1B). We confirmed that ROFA exposure of mothers on the control diet aggravated the response of their neonatal mice as assessed by mucin score and gene expressions of IL-5 and IL-13 (S2 Fig). Only ROFA exposed mice were used later and this was the focus in this study (Fig 1C).

### *B. breve* M-16V attenuates lung allergic inflammation in offspring from ROFA-exposed mothers

Next, we examined whether *B. breve* M-16V prevented allergic inflammation accelerated by ROFA. Histological evaluation of neonatal OVA-exposed lungs from mothers exposed to ROFA during pregnancy revealed a reduction of pulmonary inflammation in the M-16V diet group compared with the control diet group (Fig 2A). Furthermore, mucus levels in the airways demonstrated a significant decrease in overall mucus production in the M-16V diet group (Fig 2B).

### Expressions of allergic response genes are decreased in OVA-allergic M-16V diet mice

To elucidate the mechanism underlying the changes in pulmonary inflammation between the control and M-16V diets, we evaluated the profile of inflammatory genes in the lungs. Gene expression levels of IL-5, IL-13, and Muc5ac were significantly lower in the OVA-allergic M-16V diet group compared with the control group (Fig 3B–3D). In addition, IL-4 gene expression tended to be decreased in the OVA-allergic M-16V diet group, although there was no statistically significant difference in IL-4 level between the groups (Fig 3A).

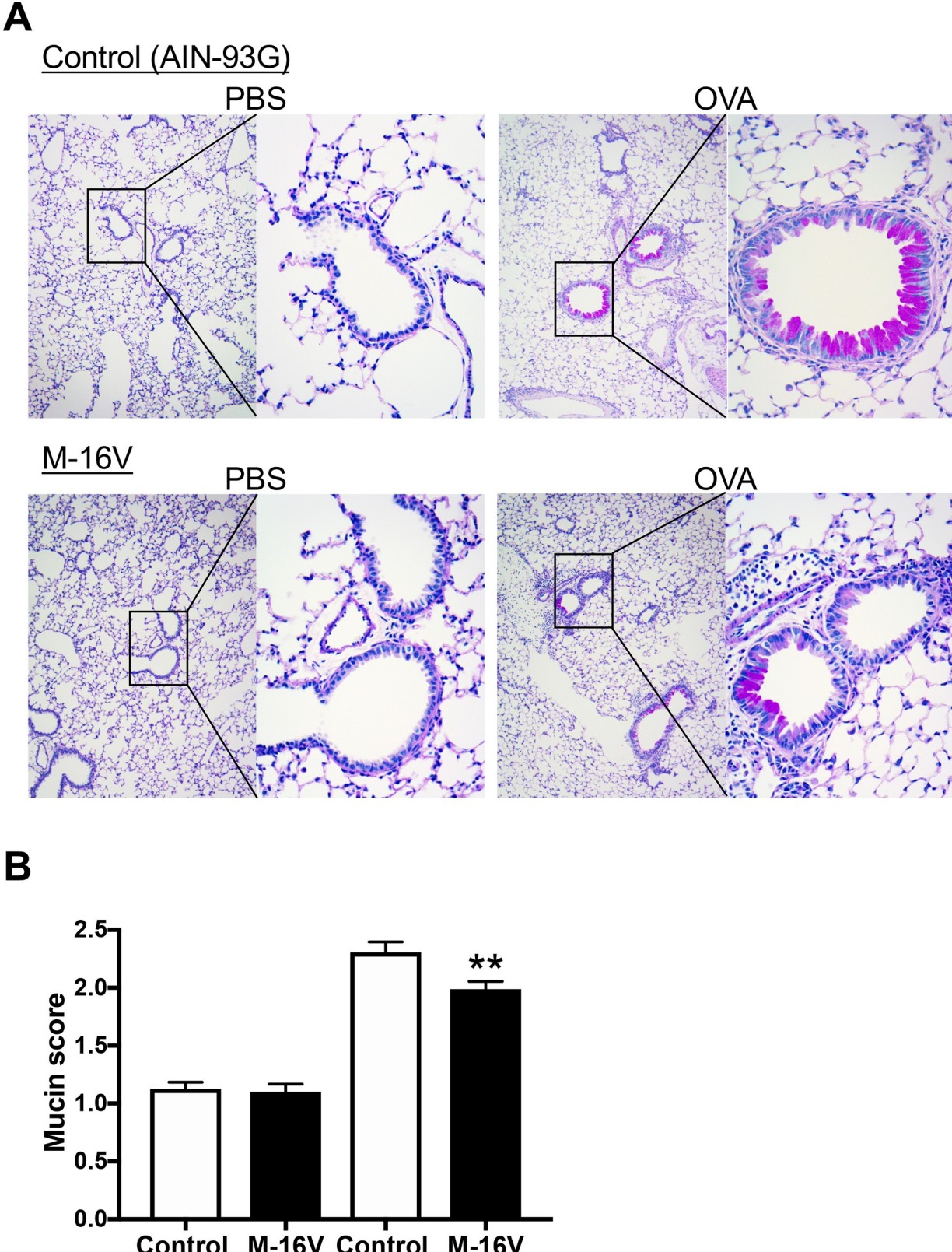

**Fig 2. Histopathologic analysis of lung tissue.** (**A**) Lung sections were stained with PAS to visualize mucus production at day 30 of life. (**B**) Quantitative analysis of mucus production using a scoring system of 1–4 detailed in the Materials and methods. The values are presented as the means ± SEM (n = 6–21). *P < 0.05 compared with OVA-sensitized and -exposed mice with a control diet.

### *B. breve* M-16V reduces eosinophil influx into airways

We also investigated the numbers of macrophages, lymphocytes, neutrophils, and eosinophils in the BALF. The number of eosinophils in the BALF was significantly decreased in the OVA-allergic M-16V diet group compared with the OVA-allergic control diet group, and there were no significant differences in the numbers of macrophages, neutrophils, and lymphocytes in OVA-exposed mice between the control and M-16V diet groups (Fig 4). Additionally, we measured cytokine

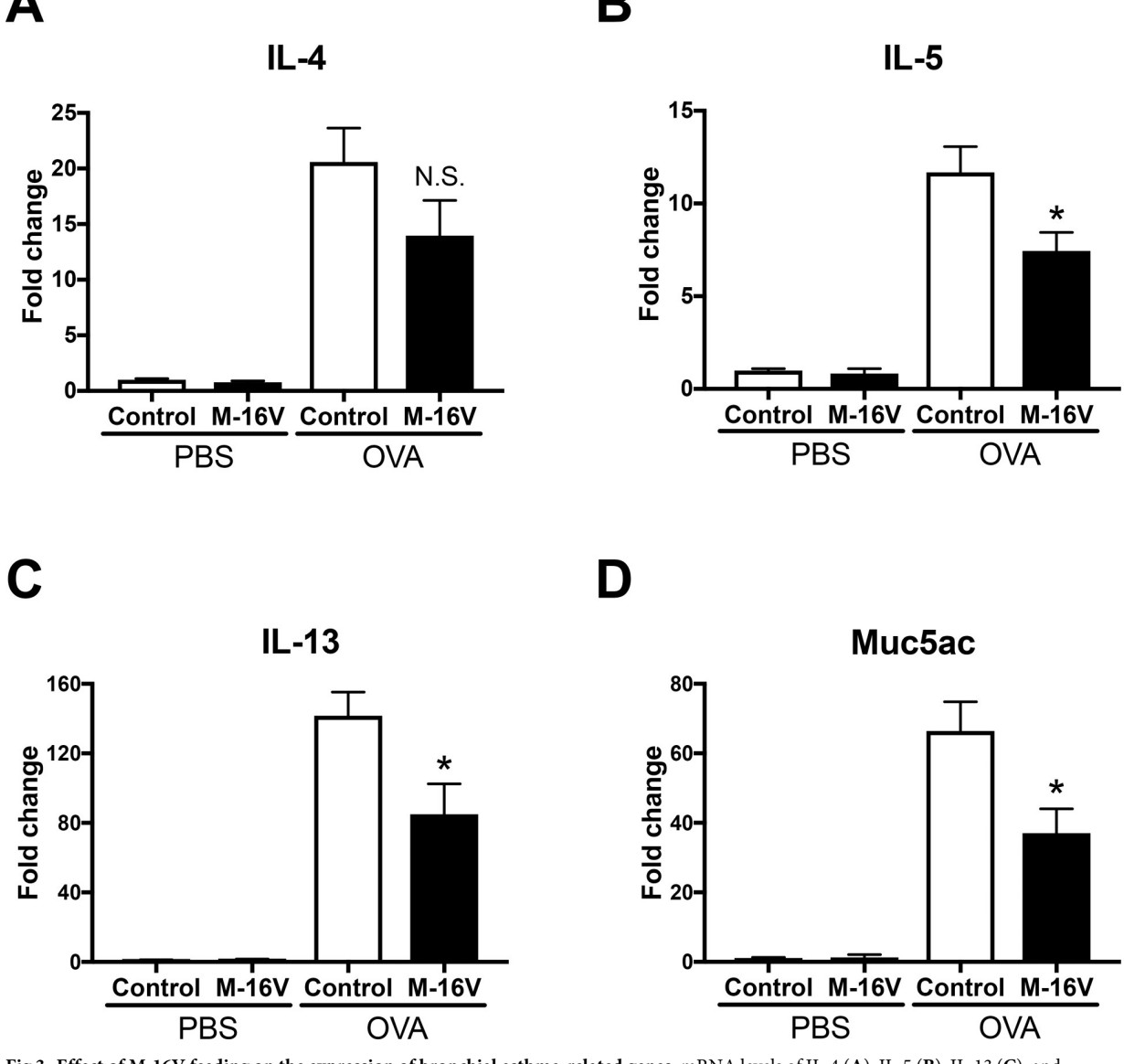

**Fig 3. Effect of M-16V feeding on the expression of bronchial asthma-related genes.** mRNA levels of IL-4 (**A**), IL-5 (**B**), IL-13 (**C**), and Muc5ac (**D**) in lungs were determined by real-time PCR. The values are presented as the means ± SEM (n = 6–21). *P < 0.05, **P < 0.01, compared with OVA-sensitized and -exposed mice with a control diet.

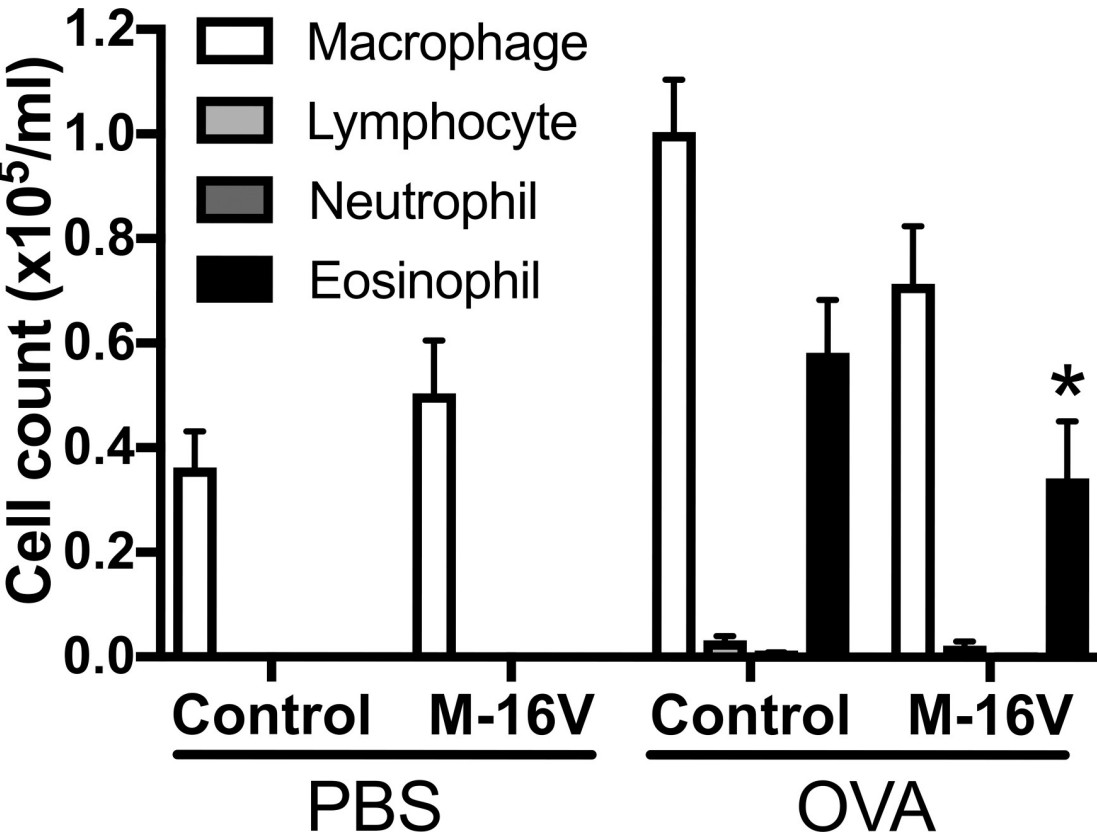

**Fig 4. Effect of M-16V diet on the number of immune cells (macrophages, lymphocytes, neutrophils, and eosinophils) in the BALF.** BALF was collected 48 h after the final aerosol challenge. The values are presented as the means ± SEM (n = 6–21). *P < 0.05 compared with OVA-sensitized and -exposed mice with a control diet.

production from the BALF. Although the gene expressions of IL-5 and IL-13 in the lungs were significantly lower in the OVA-allergic M-16V diet group compared with controls, no significant differences were found in IL-4, IL-5, or IL-13 protein production from the BALF (S3 Fig).

### *B. breve* M-16V diet changes the fecal microbiota profile

The M-16V diet prevented OVA-induced allergic inflammation in the lungs, and therefore we assessed the effect of M-16V on the fecal microbiota in neonatal mice. We compared the fecal microbiota composition of samples from neonatal mice with or without M-16V diet between control and OVA-allergic neonatal mice. The proportion of Actinobacteria was higher in the OVA-allergic M-16V diet group compared with the OVA-allergic control diet group, although there was no statistically significant difference (Fig 5A). No significant difference in the proportions of Bacteroides and Proteobacteria was observed in fecal samples between the control and *B. breve* M-16V diet groups (Fig 5B and 5C). However, the portion of Firmicutes was significantly lower in the *B. breve* M-16V diet group compared with the control group (Fig 5D).

### Specific microbial taxa in feces are associated with allergy-related factors in lungs

Next, we identified which types of fecal microbiota were regulated by the *B. breve* M-16V diet and were associated with the pathogenesis of lung allergic responses in OVA-allergic mice.

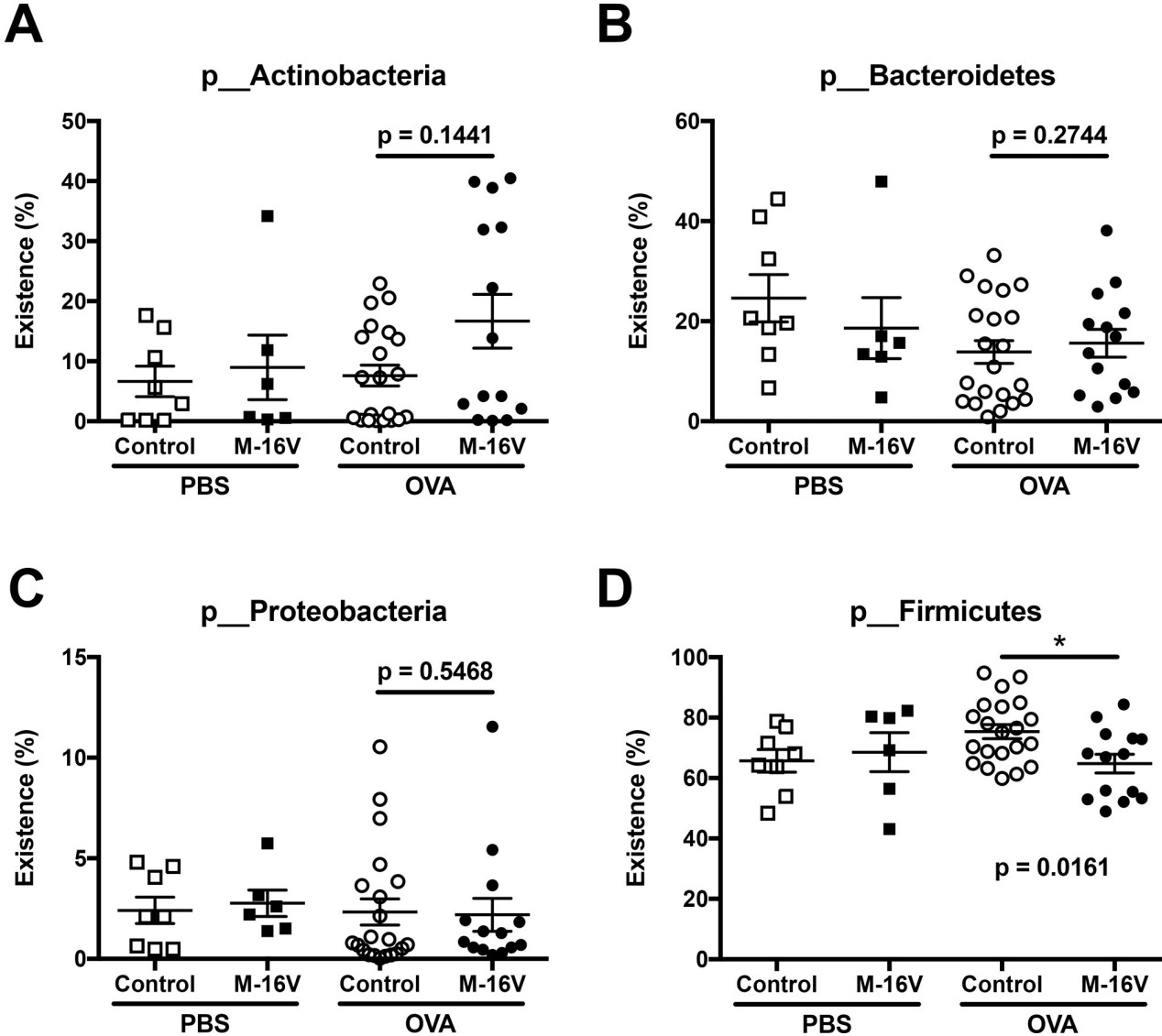

**Fig 5. Proportion of total bacteria species in feces between control and *B. breve* M-16V diet groups.** The abundances of (**A**) Actinobacteria, (**B**) Bacteroides, (**C**) Proteobacteria, and (**D**) Firmicutes in fecal samples from each mouse were analyzed by 16S rRNA gene sequencing. The values are presented as the means ± SEM (n = 6–21). *$P < 0.05$ compared between control and *B. breve* M-16V diet OVA-allergic model mice.

Using next-generation sequencing, several genera were associated with bronchial asthma-related factors such as lung histology and Th2 cytokines. In detail, *Gemella* and *Streptococcus* genera were significantly increased by the intake of *B. breve* M-16V in OVA-allergic mice, and these genera were negatively correlated with histological mucin score (Fig 6A–6D). In addition, the gene expression of the Th2 cytokine IL-5 was significantly correlated with the genera of *Lactobacillus* and *Dehalobacterium*. The *B. Breve* M-16V diet decreased the proportion of *Lactobacillus* and this had a positive correlation with IL-5 gene expression in the lungs. However, the *B. Breve* M-16V diet increased the proportion of *Dehalobacterium* and this had a negative correlation with IL-5 gene expression in the lungs (Fig 7A–7D). Moreover, gene expression of the Th2 cytokine IL-13 was also significantly correlated with two genera, *Blautia* and *Faecalibacterium*. While the *B. Breve* M-16V diet did not induce a significant proportion of *Blautia* or *Faecalibacterium*, both genera showed a negative correlation with IL-13 gene

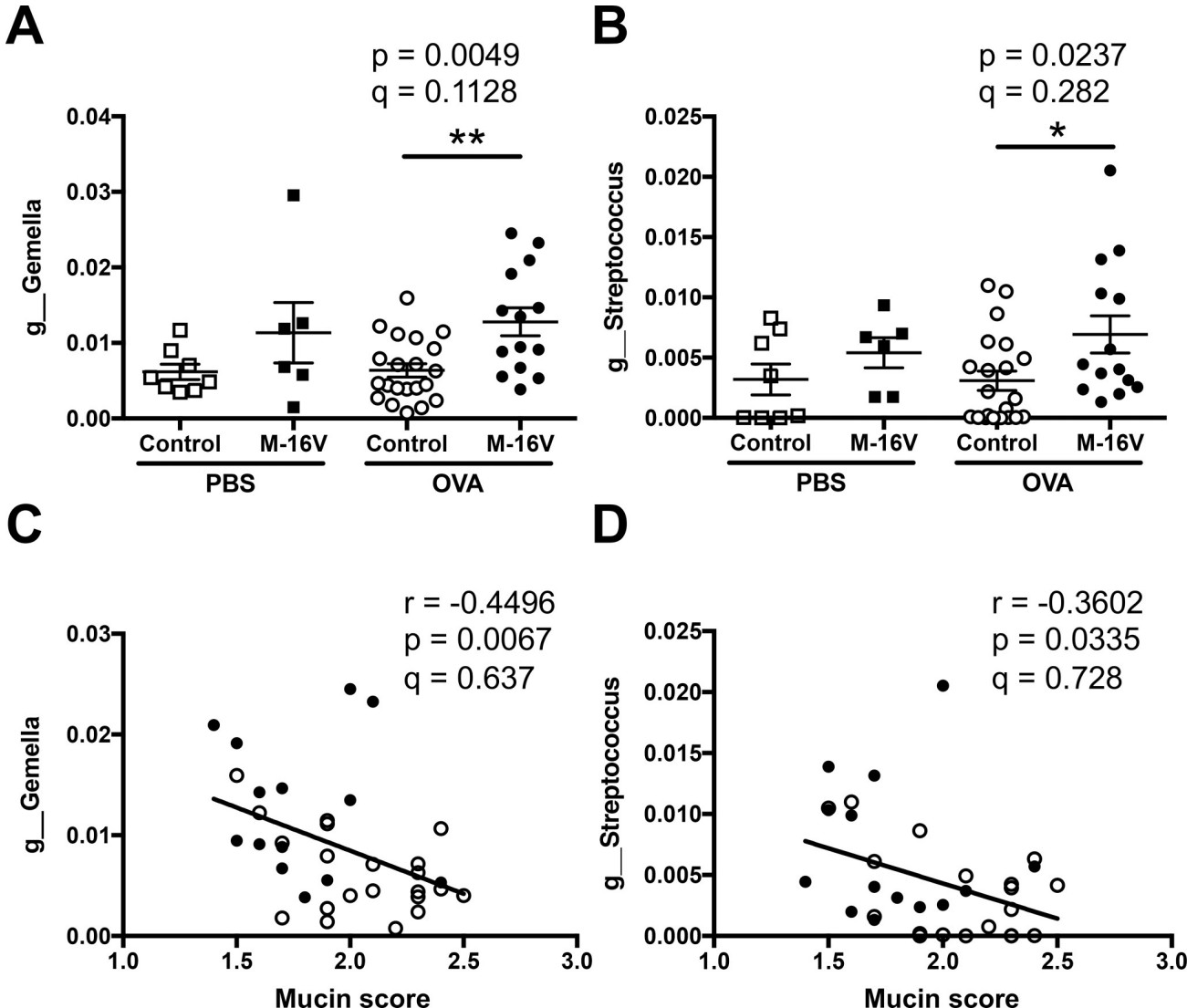

**Fig 6. Relationship of mucin score in lung histology and fecal microbiota composition of each sample.** Relative abundance of (**A**) *g_Gemella* and (**B**) *g_Streptococcus* in samples from individual mice. The values are presented as the means ± SEM (n = 6–21). *P < 0.05, **P < 0.01 compared between control and *B. breve* M-16V diet OVA-allergic model mice. Spearman's rank correlation between the relative abundance of (**C**) *g_Gemella* or (**D**) *g_Streptococcus* and mucin score shown in Fig 2B. Q-value is an FDR adjusted p-value.

expression in the lungs (Fig 8A–8D). IL-5 expression level was not correlated with *Blautia* species and the IL-13 expression level was not correlated with *Lactobacillus* species. No microbiota genus was correlated with the gene expressions of IL-4 or Muc5ac.

## Discussion

In the present study, we found that prenatal and postnatal supplementation with the *B. breve* M-16V combination was effective at reducing allergic inflammation in the airways of infant mice with OVA-induced allergic asthma delivered from a ROFA-exposed mother. Maternal supplementation with *B. breve* M-16V during pregnancy and breastfeeding changed the composition of gut microbiota; some genera correlated with several features of asthma including mucin score and Th2 cytokines including IL-5 and IL-13 in lungs.

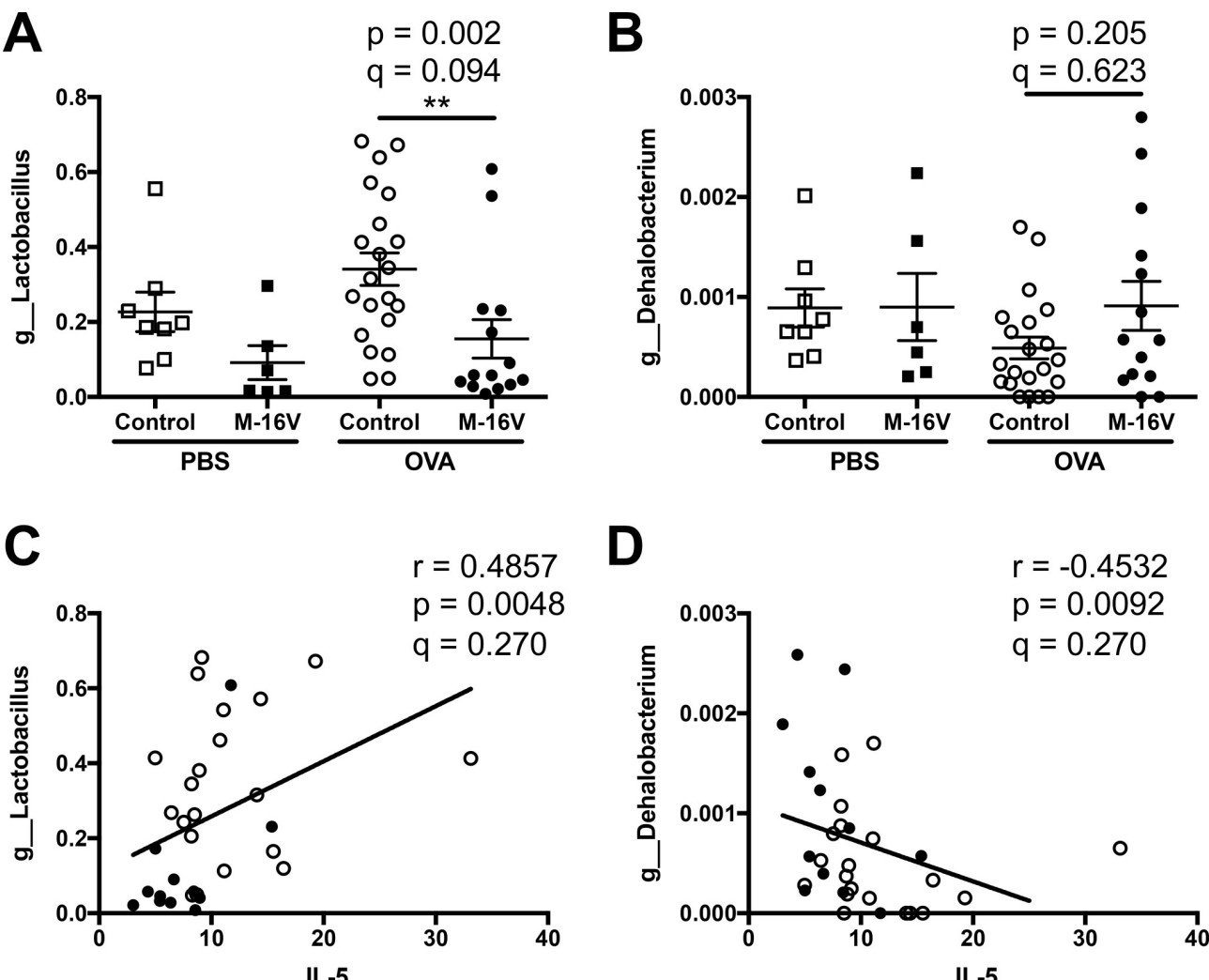

**Fig 7. Relationship between IL-5 expression in the lungs and fecal microbiota composition of each sample.** Relative abundance of (**A**) *g_Lactobacillus* and (**B**) *g_Dehalobacterium* in samples from individual mice. The values are presented as the means ± SEM (n = 6–21). \*\**P* < 0.01 compared between control and *B. breve* M-16V diet OVA-allergic model mice. Spearman's rank correlation between the relative abundance of (**C**) *g_Lactobacillus* or (**D**) *g_Dehalobacterium* and IL-5 mRNA expression shown in Fig 3B. Q-value is an FDR adjusted p-value.

Experimental and epidemiological studies have shown that air pollution factors, such as ROFA and DEP, are powerful adjuvants that promote allergic-type immune responses, which leads to skewed Th2 responses suggesting the transgenerational transmission of asthma risk after exposure to environmental particles during pregnancy [6]. In this study, the ROFA leachate contained high concentrations of metal elements, especially Ni, V, Zn, Co, Mn, Ca, and Cu [11]. Several of these metals were demonstrated to affect airway inflammation and physiologic responses in animal and epidemiologic studies [4, 35]. We previously demonstrated that the exposure of pregnant mice to an air pollutant surrogate (aerosolized ROFA leachate) exacerbated asthma-like airway Th2 inflammation in their offspring [11]. Furthermore, the maternal transfer of metals stimulated allergic responses and nickel may act as a hapten to induce allergic diseases [36]. Accordingly, the contribution of metals to ROFA-induced pulmonary injury during pregnancy was proven to be associated with increased immune cell infiltration into the lungs of offspring. However, in our previous study, aerosol exposure using a nickel sulfate

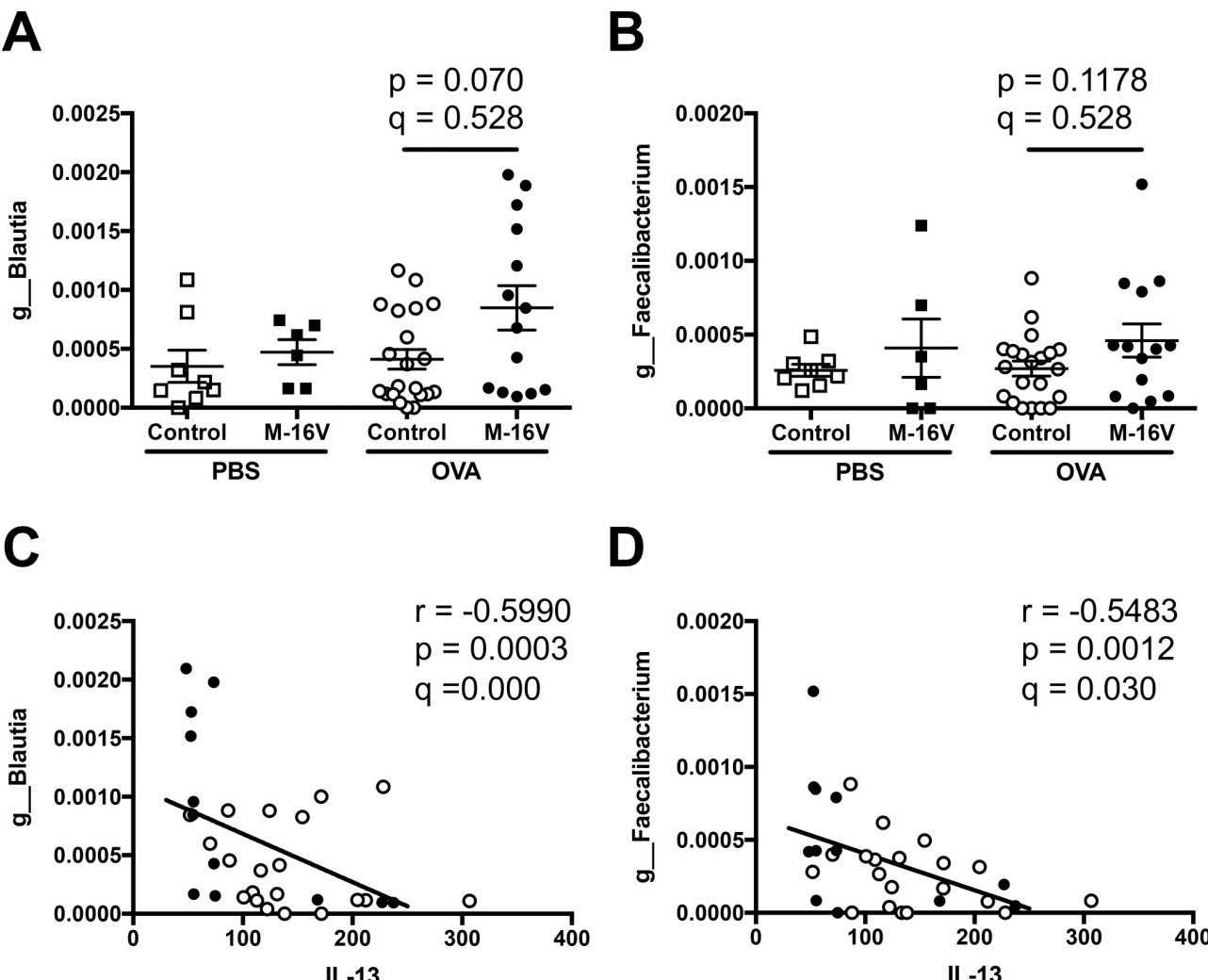

**Fig 8. Relationship between IL-13 expression in the lungs and fecal microbiota composition of each sample.** Relative abundance of (**A**) *g_Blautia* and (**B**) *g_Faecalibacterium* in samples from individual mice. The values are presented as the means ± SEM (n = 6–21). *$P < 0.05$ compared between control and *B. breve* M-16V diet OVA-allergic model mice. Spearman's rank correlation between the relative abundance of (**C**) *g_Blautia* or (**D**) *g_Faecalibacterium* and IL-13 mRNA expression shown in Fig 3C. Q-value is an FDR adjusted p-value.

solution at the same concentration and pH as the ROFA leachate did not reproduce this pro-asthmatic reaction [11], suggesting complex interactions between metals in the ROFA leachate might be required.

Probiotics are defined as live microorganisms, which, when administrated in adequate amounts, confer a health benefit to the host (FAO/WHO, 2002). Lactobacilli and bifidobacteria are the best-known probiotic candidates. Bifidobacteria are the major components of the human intestinal microbiota. Evidence is accumulating that specific probiotics including bifidobacteria counterbalance proallergic Th2-skewed immune responses [17, 37]. *B. breve* is a dominant bifidobacteria in the infant microbiota [38]. Strain M-16V, isolated from the fecal sample of a human baby [20], promoted the colonization of bifidobacteria and the formation of normal intestinal microbiota in low-birth-weight infants [22]. Furthermore, Hougee *et al.* demonstrated that, among six bacterial strains with potent antiallergic effects including four *Bifidobacterium* strains and two *Lactobacillus* strains, *B. breve* M-16V was the most effective at reducing allergic responses [39]. Here, we demonstrated that the oral administration of M-

16V to mice during prenatal and postnatal periods improved asthma allergic inflammation in their offspring. Probiotics, especially M-16V, might facilitate the development of gut immune function and attenuate inflammation in preterm infants by changing gut microbial colonization [22]. Although recent studies have suggested that maternal supplementation with probiotics during pregnancy and breastfeeding may reduce the risk of eczema in their infants [40, 41], prenatal intervention alone did not affect the risk of eczema in infants [42]. According to this evidence, we chose maternal supplementation with probiotics during pregnancy and breastfeeding. Further studies are needed to evaluate the effect of M-16V supplementation in mothers on their offspring by nursing alone.

By analyzing the fecal microbiota composition of samples from neonatal mice, we found a lower proportion of Firmicutes in the OVA-allergic M-16V diet group compared with the OVA-allergic control diet group. A detailed analysis of the gut microbiota showed that *Gemella* was increased in the OVA-allergic M-16V diet group. Furthermore, this study showed it was inversely related to the mucin score. Sjödin *et al.* demonstrated that the inoculation of germ-free mice with *Gemella* decreased airway inflammation in their offspring [43]. *Dehalobacterium* was reported to be a key driver of sex-specific gut microbiota profiles [44], but its role in allergic responses is still unknown. The increased proportion of *Streptococcus* and *Blautia* may be negative regulators of the mucin score and IL-13, respectively. It is reported that *Streptococcus* and *Blautia* produce substances such as bacteriocins, enzymes, lactic acid, and fatty acids [45]. Furthermore, few bacterial genera, such as most members of *Akkermansia*, *Allobaculum*, *Bacteroides*, and *Blautia* produce short-chain fatty acids [46]. Previous studies reported that short-chain fatty acids reduced the risk of allergic airway disease and food allergy in offspring [47, 48]. Moreover, a high level of *Faecalibacterium* was strongly associated with a decreased level of IL-13. A decreased relative abundance of *Faecalibacterium* in early infancy was reported to be associated with increased asthma risk [44, 49]. These data suggest that in our study, higher levels of *Blautia* and *Faecalibacterium* had a strong relationship with decreased IL-13 expression in the lungs, which may be correlated with the reduced allergic responses by *B. breve* M-16V. Further investigation is required to determine the association between allergic responses and the microbiota and between the microbiota and each allergic-related parameter.

The pathogenesis of allergic asthma involves Th2-type cytokines, such as IL-4, IL-5, and IL-13 [13, 50]. IL-4 is a key cytokine that drives type-2 responses, IL-5 recruits eosinophils, and IL-13 is a pleiotropic cytokine produced predominantly by Th2 cells [51]. In our model, the gene expression levels of IL-5 and IL-13 in the lungs were significantly inhibited while IL-4 expression was intact in the OVA-allergic M-16V diet group compared with the control group. Moreover, the mucin score and expression of Muc5ac, which is localized to goblet cells in the surface epithelium and contributes to mucus production that correlates with asthma pathogenesis [52], were significantly improved in the M-16V diet group. Our previous report indicated that ROFA exposure in mothers did not enhance IL-4 production in OVA-allergic neonatal mice [11]. Additionally, recent accumulating studies demonstrated that type 2 innate lymphoid cells (ILC2) have a critical role in the pathogenesis of allergic asthma [53]. ILC2 mainly produced IL-5 and IL-13, but not IL-4, in an allergic asthma model [54]. Therefore, ILC2 might be suppressed by the administration of M-16V in our model. This evidence supports our data, which indicated that the gene expression of IL-4 was not significantly changed by the M-16V diet. We also measured protein levels (IL-4, IL-5, and IL-13) in BALF, however, there was no significant difference in these cytokines. We expect deviation of time course between gene expressions in lungs and cytokine productions in BALF. Further investigation including the role of ILC2 and the time point of analysis in the M-16V diet in allergic inflammation is necessary.

In summary, we demonstrated that the administration of *B. breve* M-16V to mothers during pregnancy and breastfeeding prevented the development of OVA-induced asthma-like allergic inflammation accelerated by air pollution ROFA in their offspring by inhibiting Th2 immune responses and changes in their gut microbiota. Of note, *B. breve* M-16V is safe for use as a probiotic in humans including premature infants and pregnant women [23]. Thus, additional studies of the preventive potential of M-16V for allergic asthma are warranted, and M-16V might ultimately have clinical applicability.

## Supporting information

**S1 Fig. Lung histopathologic analysis of OVA allergic neonates from mothers exposed to PBS (no ROFA) on a control (AIN-93G) or M-16V diet.** (**A**) Lung sections were stained with PAS to visualize mucus production at day 30 of life. (**B**) Quantitative analysis of mucus production using a scoring system of 1–4 detailed in the Materials and methods. The values are presented as the means ± SEM (n = 6–8).
(TIF)

**S2 Fig. Lung histopathologic analysis of OVA allergic neonates from PBS- or ROFA-exposed mothers on a control (AIN-93G) diet.** (**A**) Lung sections were stained with PAS to visualize mucus production at day 30 of life. (**B**) Quantitative analysis of mucus production using a scoring system of 1–4 detailed in the Materials and methods. The values are presented as the means ± SEM (n = 6–7). $^*P < 0.05$ compared with OVA-sensitized and -exposed neonates from mothers without ROFA exposure.
(TIF)

**S3 Fig. Cytokine production from the BALF supernatant.** BALF was collected 48 h after the final aerosol challenge, and BALF supernatant was collected after the centrifugation of samples at 800 ×*g* for 5 min. Cytokine productions of IL-4, IL-5, and IL-13 from BALF supernatants were measured by ELISA. The values are presented as the mean ± SEM (n = 6–21).
(TIF)

## Acknowledgments

We thank Ms. Reiko Masuda, Ms. Hisayo Nishikawa, and Mr. Ryo Misawa (Nara Medical University) for their assistance. We thank J. Ludovic Croxford, PhD, from Edanz Group (https://en-author-services.edanzgroup.com/) for editing a draft of this manuscript.

## Author Contributions

**Conceptualization:** Chiyoko Terada-Ikeda, Masahiro Kitabatake, Noriyuki Iwabuchi, Toshihiro Ito.

**Data curation:** Chiyoko Terada-Ikeda, Masahiro Kitabatake, Noriko Ouji-Sageshima, Toshihiro Ito.

**Formal analysis:** Chiyoko Terada-Ikeda, Masahiro Kitabatake, Kumiko Kato, Toshihiro Ito.

**Funding acquisition:** Toshihiro Ito.

**Investigation:** Chiyoko Terada-Ikeda, Masahiro Kitabatake, Satsuki Yasui, Natsuko Imakita, Noriko Ouji-Sageshima, Toshihiro Ito.

**Methodology:** Chiyoko Terada-Ikeda, Masahiro Kitabatake, Toshihiro Ito.

**Project administration:** Toshihiro Ito.

**Resources:** Akari Hiraku, Kaoru Hamada, Toshihiro Ito.

**Software:** Chiyoko Terada-Ikeda, Masahiro Kitabatake, Kumiko Kato.

**Supervision:** Noriyuki Iwabuchi, Kaoru Hamada, Toshihiro Ito.

**Validation:** Chiyoko Terada-Ikeda, Masahiro Kitabatake, Akari Hiraku, Kumiko Kato, Noriko Ouji-Sageshima, Noriyuki Iwabuchi, Toshihiro Ito.

**Visualization:** Chiyoko Terada-Ikeda, Masahiro Kitabatake, Akari Hiraku, Noriko Ouji-Sageshima, Toshihiro Ito.

**Writing – original draft:** Masahiro Kitabatake, Toshihiro Ito.

**Writing – review & editing:** Chiyoko Terada-Ikeda, Masahiro Kitabatake, Akari Hiraku, Kumiko Kato, Noriko Ouji-Sageshima, Noriyuki Iwabuchi, Kaoru Hamada, Toshihiro Ito.

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
