## [Decision Letter · Decision Letter 0]

20 Apr 2020

PONE-D-20-05759

Bifidobacterium breve M-16V prevents allergic airway inflammation accelerated by prenatal exposure to an air pollutant aerosol

PLOS ONE

Dear Dr. Ito,

Thank you for submitting your manuscript to PLOS ONE. After careful consideration, we feel that it has merit but does not fully meet PLOS ONE’s publication criteria as it currently stands. Therefore, we invite you to submit a revised version of the manuscript that addresses the points raised during the review process.

Two experts from the field and I myself have reviewed your manuscript. Although the study is in general interesting and well performed, there are several points that need to be clarified in a revised version of the manuscript. In particular, but not excluively, the lack of control groups not being exposed to ROFA would have been desirable. This has also implications for the title of your manuscript. Otherwise, the manuscript cannot make any convincing conclusion about the role of ROFA in this setting. Further, the statistics used need to be partly revised and several details should be added in a revised version of the manuscript. A detailed point-by-point response to all the comments will be mandatory.

We would appreciate receiving your revised manuscript by Jun 04 2020 11:59PM. To enhance the reproducibility of your results, we recommend that if applicable you deposit your laboratory protocols in protocols.io, where a protocol can be assigned its own identifier (DOI) such that it can be cited independently in the future. For instructions see: http://journals.plos.org/plosone/s/submission-guidelines#loc-laboratory-protocols

We look forward to receiving your revised manuscript.

Kind regards,

Heinz Fehrenbach

Academic Editor

PLOS ONE

Journal Requirements:

2. We note that you are reporting an analysis of a microarray, next-generation sequencing, or deep sequencing data set. PLOS requires that authors comply with field-specific standards for preparation, recording, and deposition of data in repositories appropriate to their field. Please upload these data to a stable, public repository (such as ArrayExpress, Gene Expression Omnibus (GEO), DNA Data Bank of Japan (DDBJ), NCBI GenBank, NCBI Sequence Read Archive, or EMBL Nucleotide Sequence Database (ENA)). In your revised cover letter, please provide the relevant accession numbers that may be used to access these data. For a full list of recommended repositories, see http://journals.plos.org/plosone/s/data-availability#loc-omics or http://journals.plos.org/plosone/s/data-availability#loc-sequencing.

3. Thank you for stating the following in the Financial Disclosure section: "The author(s) received no specific funding for this work."

We note that one or more of the authors are employed by a commercial company: 'Morinaga Milk Industry Co., Ltd'.

Additional Editor Comments (if provided):

In additional to the reviewers' comments, I raise the following points:

- the effects in the OVA models after ROFA exposure during pregancy are very mild; so if ROFA aggravates the response of neonatal mice to OVA, I wonder whtethger there is any significant effect in mice that were not exposed to ROFA

- only few cytokines were analysed / presented and only at the transcriptional level; it would be important to also give the protein levels in BALF

- IL-4 is of majopr importance for eosinophilc inflammation; however, expression of IL-4 was not significantly changed by the diet; please comment

- although the methodology of mucin scoring was referenced, it would be helpful to briefly state how the analysis was performed and statistically analysed

- histopathological overview micrographs should be complemented by high power micrographs to be able to assess the inflammaion

Reviewers' comments:

Reviewer's Responses to Questions

**Comments to the Author**

1. Is the manuscript technically sound, and do the data support the conclusions?

Reviewer #1: Yes

Reviewer #2: Partly

2. Has the statistical analysis been performed appropriately and rigorously? 

Reviewer #1: Yes

Reviewer #2: No

3. Have the authors made all data underlying the findings in their manuscript fully available?

Reviewer #1: Yes

Reviewer #2: No

4. Is the manuscript presented in an intelligible fashion and written in standard English?

Reviewer #1: Yes

Reviewer #2: Yes

5. Review Comments to the Author

Reviewer #1: I have following recommendations and comments on the manuscript with the title "Bifidobacterium breve M-16V prevents allergic airway inflammation accelerated by prenatal exposure to an air pollutant aerosol":

1. The relevance of exposing the pregnant mice with ROFA in the mouse model is hardly comprehensible. Data from mice which were not ROFA exposed is not included. It seems that the Th2 immune reaction in the offspring is induced by OVA exposure rather than additional ROFA treatment of the mothers. The authors should explicitely figure out why ROFA exposure of the pregnant animals is of importance for their manuscript. If possible, they should include a short comparative data set of Th2 inflammation markers of OVA-exposed offsprings from ROFA and not ROFA +/- M-16V treated mothers.

2. In the description of the mouse model it´s unclear if M-16V diet of the mothers was continued throughout the nursing time. If so, the authors should at least shortly discuss a possible effect of M-16V on the offsprings just by nursing.

3. In figure 6 and 7 correlation calculations were performed on the incidence of bacterial species with Th2 cytokine mRNA expression. However, the information is missing if or not IL-5 expression level is also correlated e.g. with Blautia species and IL-13 e.g. with Lactobaccillus spec.

4. In figure 2 and 5 please correct the wording "mucin score" at the Y- and X-axis, respectively.

Reviewer #2: In the present manuscript titled „ Bifidobacterium breve M-16V prevents allergic airway inflammation accelerated by prenatal exposure to an air pollutant aerosol” Terada-Ikeda et al examined the effect of the bifidobacterial strain M-16V fed to pregnant mice on the severity of allergic airway disease in offspring mice.

The study is well structured, concise and contains interesting findings. However, I have several major points regarding the experimental-setup.

X the title is misleading as it gives emphasis on pollution expose of mothers, which was not investigated in this study - all animals were treated with ROFA. I understand that a ROFA-treated group stands on its own, however, a control group (no ROFA exposure) would have largely benefited the study, and authors should discuss why no such group was included. In line with this, the first page of the discussion section can be omitted as it deals in detail with the effects of ROFA treatment.

X the samples sizes applied are not clear. They are not given in the materials and methods section. In the Figure legends they vary between parameters, namely, n=4-8 in Fig 1-3 and n=6-21 in Figures displaying results from microbiota analyses. Why is there such a difference in sample sizes? It seems not all parameters were measured for all animals?

X How were mice housed? In single cages or in groups? For microbiota studies cage-effects are widely known.

X stats applied are not always appropriate. It reads that ANOVA was used throughout, however, often microbiota data is not normally distributed (see e.g. Streptococcus). Furthermore, correction for multiple testing has to be applied.

X what was the diet of neonates? Initially, I thought they were breast-fed, however, reading the first sentence in the discussion section I was confused as authors speak about postnatal supplementation of the probiotic. Nothing is given in the material and methods section.

X more information on library preparations should be given. Previous work is referenced, yet major information should be included here as well, such as the kit used for DNA extraction, what primers were used, … .

X no link for 16S rRNA-gene data download is given.

Minor points

X what was the sex of offsprings?

X L263: p-value for Actinos = 0.14, so it is not significant.

X L265: according to Fig 5 Gemella comprises only ~1% of total bacteria, so it is not predominant.

X L266: mucus microbiota was not investigated, but fecal samples were used for analyses.

X 273: Blautia is not a major butyrate producer in gut microbiota – only some specific strains might produce this compound.

X 280: omit “led-to” – it implies causality, which was not investigated here. In general, care should be taken with the wording throughout the study as only associations were obtained.

X some parts of the discussion read like a commercial for M-16V.

6. PLOS authors have the option to publish the peer review history of their article (what does this mean?). If published, this will include your full peer review and any attached files.

Reviewer #1: No

Reviewer #2: No

---

## [Author Response · Author response to Decision Letter 0]

17 Jul 2020

In additional to the reviewers' comments, I raise the following points:

We do appreciate the comments and suggestions by the editor, and we have addressed each with additional comments and experiments.

- the effects in the OVA models after ROFA exposure during pregnancy are very mild; so if ROFA aggravates the response of neonatal mice to OVA, I wonder whether there is any significant effect in mice that were not exposed to ROFA.

Here, we first assessed whether ROFA aggravates the response of neonatal mice to OVA. We confirmed that ROFA exposure in mothers aggravates the response of neonatal mice by mucin score and gene expressions of IL-5 and IL-13 (S2 Fig). We have also previously reported that offspring from mothers exposed to ROFA during pregnancy induced robust pathologic changes of airway inflammation including mucin scoring (Ref 11).

Moreover, we compared the response of neonatal mice to OVA without ROFA exposure between control (AIN-93G) and M-16V diet. As shown in S1 Fig, there was no significant effect in mice that were not exposed to ROFA between control (AIN-93G) and M-16V diet. 

- only few cytokines were analyzed / presented and only at the transcriptional level; it would be important to also give the protein levels in BALF

We measured protein levels (IL-4, IL-5, and IL-13) in BALF. However, there was no significant difference in these cytokines (S3 Fig). We expect deviation of time course between gene expressions in lungs and cytokine productions in BALF. We mentioned it in “Discussion” (L364-367). 

- IL-4 is of major importance for eosinophilc inflammation; however, expression of IL-4 was not significantly changed by the diet; please comment.

We totally agree that IL-4 is one of major Th2 cytokines including eosinophilic inflammation. Our previous report indicated that ROFA exposure in mothers did not enhance IL-4 production in OVA-allergic neonatal mice (Ref 11). Here, we also demonstrate that ROFA exposure in mothers aggravated the gene expressions of IL-5 and IL-13, not IL-4, in lungs of neonatal mice (S2 Fig), and that those of IL-5 and IL-13, not IL-4, were significantly lower in the OVA-allergic M-16V diet group compared with the control diet (AIN-93G) group. Also, there was no significant effect in mice that were not exposed to ROFA between control (AIN-93G) and M-16V diet (S1 Fig). 

Additionally, recent accumulating studies have demonstrated that type 2 innate lymphoid cells (ILC2) have a critical role in the pathogenesis of allergic asthma (Ref. 53). ILC2 mainly produced IL-5 and IL-13, but not IL-4 in an allergic asthma model (Ref. 54). This evidence supports our data, which indicated that the gene expression of IL-4 was not significantly changed by the M-16V diet. We added the above comment to “Discussion” (L357-368). 

- although the methodology of mucin scoring was referenced, it would be helpful to briefly state how the analysis was performed and statistically analyzed.

 The methodology of mucin scoring was mentioned in “Materials and methods” (L125-128).

- histopathological overview micrographs should be complemented by high power micrographs to be able to assess the inflammation.

 High power histological micrographs were added to Figure 2A.

Reviewer #1: 

I have following recommendations and comments on the manuscript with the title "Bifidobacterium breve M-16V prevents allergic airway inflammation accelerated by prenatal exposure to an air pollutant aerosol":

We do appreciate the comments and suggestions by Reviewer #1, and we have addressed each with additional comments and experiments.

1. The relevance of exposing the pregnant mice with ROFA in the mouse model is hardly comprehensible. Data from mice which were not ROFA exposed is not included. It seems that the Th2 immune reaction in the offspring is induced by OVA exposure rather than additional ROFA treatment of the mothers. The authors should explicitely figure out why ROFA exposure of the pregnant animals is of importance for their manuscript. If possible, they should include a short comparative data set of Th2 inflammation markers of OVA-exposed offsprings from ROFA and not ROFA +/- M-16V treated mothers.

We totally agree that our previous manuscript was not enough to explain why ROFA exposure of the pregnant animals is of importance. Here, we first assessed whether ROFA aggravates the response of neonatal mice to OVA. We confirmed that ROFA exposure in mothers aggravates the response of neonatal mice by mucin score and gene expressions of IL-5 and IL-13 (S1 Fig). We have also previously reported that offspring from mothers exposed to ROFA during pregnancy induced robust pathologic changes of airway inflammation including mucin scoring (Ref 11).

Moreover, we compared the response of neonatal mice to OVA without ROFA exposure between control (AIN-93G) and M-16V diet. As shown in S2 Fig, there was no significant effect in mice that were not exposed to ROFA between control (AIN-93G) and M-16V diet. 

2. In the description of the mouse model it´s unclear if M-16V diet of the mothers was continued throughout the nursing time. If so, the authors should at least shortly discuss a possible effect of M-16V on the offsprings just by nursing.

M-16V diet of the mothers was continued throughout the nursing time until analysis of offsprings. So, we now discuss a possible effect of M-16V on the offsprings just by nursing (L325-327).

3. In figure 6 and 7 correlation calculations were performed on the incidence of bacterial species with Th2 cytokine mRNA expression. However, the information is missing if or not IL-5 expression level is also correlated e.g. with Blautia species and IL-13 e.g. with Lactobaccillus spec.

We just pick up the results which were significantly correlated with between gene expression in lungs and bacteria genera in fecus. IL-5 expression level was not correlated with Blautia species, and IL-13 expression level was not correlated with Lactobaccillus spec. It is now mentioned in “Result” (L279-281).

4. In figure 2 and 5 please correct the wording "mucin score" at the Y- and X-axis, respectively.

Thank you for pointing it out. We corrected the wording “mucin score”.

Reviewer #2: 

In the present manuscript titled „ Bifidobacterium breve M-16V prevents allergic airway inflammation accelerated by prenatal exposure to an air pollutant aerosol” Terada-Ikeda et al examined the effect of the bifidobacterial strain M-16V fed to pregnant mice on the severity of allergic airway disease in offspring mice.

The study is well structured, concise and contains interesting findings. However, I have several major points regarding the experimental-setup.

We do appreciate the comments and suggestions by Reviewer #2, and we have addressed each with additional comments and experiments.

1. The title is misleading as it gives emphasis on pollution expose of mothers, which was not investigated in this study - all animals were treated with ROFA. I understand that a ROFA-treated group stands on its own, however, a control group (no ROFA exposure) would have largely benefited the study, and authors should discuss why no such group was included. In line with this, the first page of the discussion section can be omitted as it deals in detail with the effects of ROFA treatment.

We totally agree that the title is misleading in our previous manuscript because all animals were treated with ROFA. The editor and all reviewers pointed out that there was no control group (no ROFA exposure). Here, we first assessed whether ROFA aggravates the response of neonatal mice to OVA. We confirmed that ROFA exposure in mothers aggravates the response of neonatal mice by mucin score and gene expressions of IL-5 and IL-13 (S2 Fig). We have also previously reported that offspring from mothers exposed to ROFA during pregnancy induced robust pathologic changes of airway inflammation including mucin scoring (Ref 11).

Moreover, we compared the response of neonatal mice to OVA without ROFA exposure between control (AIN-93G) and M-16V diet. As shown in S1 Fig, there was no significant effect in mice that were not exposed to ROFA between control (AIN-93G) and M-16V diet. 

2. The samples sizes applied are not clear. They are not given in the materials and methods section. In the Figure legends they vary between parameters, namely, n=4-8 in Fig 1-3 and n=6-21 in Figures displaying results from microbiota analyses. Why is there such a difference in sample sizes? It seems not all parameters were measured for all animals?

We analyzed again the data (BAL, Real-time PCR, Mucin score) using n=6-21 in Figures displaying results from microbiota analyses. As shown in figure 2-4, we indicate new data (n=6-21), and the result trend does not change compared with previous data (n=4-8).

3. How were mice housed? In single cages or in groups? For microbiota studies cage-effects are widely known.

Each mother was house with her offspring in single cage. It was mentioned in “Materials and methods” (L90-92).

4. Stats applied are not always appropriate. It reads that ANOVA was used throughout, however, often microbiota data is not normally distributed (see e.g. Streptococcus). Furthermore, correction for multiple testing has to be applied.

The analyses of gut microbiota without the calculation of a false discovery rate (FDR) were performed using SPSS version 23.0 statistical software (IBM Corp., Armonk, NY). Intergroup differences were analyzed using the Mann-Whitney U-test and correlation analysis between gut bacteria and Th2 responses in the lungs were performed by Spearman’s correlation coefficient. Results were adjusted by FDR using the Benjamini and Hochberg method in R software version 3.6.0. (L196-202).

5. What was the diet of neonates? Initially, I thought they were breast-fed, however, reading the first sentence in the discussion section I was confused as authors speak about postnatal supplementation of the probiotic. Nothing is given in the material and methods section.

We are sorry for confusing. They were breast-fed. We now mention it in “Materials and methods” (L91).

6. More information on library preparations should be given. Previous work is referenced, yet major information should be included here as well, such as the kit used for DNA extraction, what primers were used. 

We added more information related to library preparation including DNA extraction method and primer sequencings to "Materials and methods" (L143-176).

7. No link for 16S rRNA-gene data download is given.

DNA sequences corresponding to the 16S rRNA gene data have been deposited in DDBJ under accession number DRA010221 (L186-188).

8. What was the sex of offsprings?

All offsprings (both male and female) was used. It is also mentioned in “Materials and methods” (L91).

9. L263: p-value for Actions = 0.14, so it is not significant.

We deleted the sentence about Actinobacteria.

10. L265: according to Fig 5 Gemella comprises only ~1% of total bacteria, so it is not predominant.

 We deleted the sentence together with comment #11.

11. L266: mucus microbiota was not investigated, but fecal samples were used for analyses.

 We deleted the sentence together with comment #10.

12. L273: Blautia is not a major butyrate producer in gut microbiota – only some specific strains might produce this compound.

 We agree that Blautia is not a major butyrate producer in gut microbiota, so we changed the sentence about Blautia according to Ref. 45 & 46(L337-340).

13. L280: omit “led-to” – it implies causality, which was not investigated here. In general, care should be taken with the wording throughout the study as only associations were obtained.

We omitted “led to”, and changed the sentence (L346).

14. Some parts of the discussion read like a commercial for M-16V.

We deleted some sentences which read like a commercial for M-16V in Discussion.

---

## [Decision Letter · Decision Letter 1]

11 Aug 2020

PONE-D-20-05759R1

Bifidobacterium breve M-16V prevents allergic airway inflammation accelerated by prenatal exposure to an air pollutant aerosol

PLOS ONE

Dear Dr. Ito,

Thank you for submitting your manuscript to PLOS ONE. After careful consideration, we feel that it has merit but does not fully meet PLOS ONE’s publication criteria as it currently stands. Therefore, we invite you to submit a revised version of the manuscript that addresses the points raised during the review process.

Please address the few points made by the reviewers when revising your manuscript. In particular, edit the text so that it will be clear to the reader which experiments were indeed made for which question. Revise figure 1 so that it will be clear which groups were used for which experiments. Please note that there will be no text editing step after formal acceptance of a manuscript which means that text editing always needs another round of revising a manuscript for PLoS One even though everything else may be okay.

We look forward to receiving your revised manuscript.

Kind regards,

Heinz Fehrenbach

Academic Editor

PLOS ONE

Reviewers' comments:

Reviewer's Responses to Questions

**Comments to the Author**

1. If the authors have adequately addressed your comments raised in a previous round of review and you feel that this manuscript is now acceptable for publication, you may indicate that here to bypass the “Comments to the Author” section, enter your conflict of interest statement in the “Confidential to Editor” section, and submit your "Accept" recommendation.

Reviewer #1: All comments have been addressed

Reviewer #2: (No Response)

2. Is the manuscript technically sound, and do the data support the conclusions?

Reviewer #1: Yes

Reviewer #2: Yes

3. Has the statistical analysis been performed appropriately and rigorously? 

Reviewer #1: N/A

Reviewer #2: Yes

4. Have the authors made all data underlying the findings in their manuscript fully available?

Reviewer #1: Yes

Reviewer #2: Yes

5. Is the manuscript presented in an intelligible fashion and written in standard English?

Reviewer #1: Yes

Reviewer #2: Yes

6. Review Comments to the Author

Reviewer #1: Some final comments:

1. Lines 337-339: The sentence sounds incomplete. Please check.

2. Line 347: The wording "improved allergic responses" can be misleading. Better use "reduced allergic responses".

Reviewer #2: most of my comments were adressed. However, I feel it reads a bit complicated now, so I have the following suggestions:

1. Change the title to a more simplistic one - it is hard to get the message

2. The additional first paragraphs in the results section confuse the reader on what was done in the following experiments. State clearly which mice where used for experiments and what exposure was applied. Say that only ROFA exposed mice were used later and that this was the focus.

3. Accordingly, Figure 1 is misleading as it suggests all groups have been investigated throughout the experiment - please communicate the experimental setup clearly to the reader.

7. PLOS authors have the option to publish the peer review history of their article (what does this mean?). If published, this will include your full peer review and any attached files.

Reviewer #1: No

Reviewer #2: No

---

## [Author Response · Author response to Decision Letter 1]

20 Aug 2020

Reviewer #1: 

Some final comments:

We again appreciate the comments and suggestions by Reviewer #1, and we have addressed each with additional comments.

1. Lines 337-339: The sentence sounds incomplete. Please check.

Thank you for the suggestion. The sentence was incomplete. We changed the sentence as follows; “It is reported that Streptococcus and Blautia produce substances such as bacteriocins, enzymes, lactic acid, and fatty acids.”

2. Line 347: The wording "improved allergic responses" can be misleading. Better use "reduced allergic responses".

We now use "reduced allergic responses" as suggested.

Reviewer #2: 

Most of my comments were addressed. However, I feel it reads a bit complicated now, so I have the following suggestions:

We again appreciate the comments and suggestions by Reviewer #2, and we have addressed each with additional comments.

1. Change the title to a more simplistic one - it is hard to get the message

Thank you for the suggestion. We changed the title as follows to send the message for readers; “Maternal supplementation with Bifidobacterium breve M-16V prevents their offspring from allergic airway inflammation accelerated by the prenatal exposure to an air pollutant aerosol”.

2. The additional first paragraphs in the results section confuse the reader on what was done in the following experiments. State clearly which mice where used for experiments and what exposure was applied. Say that only ROFA exposed mice were used later and that this was the focus.

We modified Figure 1 to state clearly which mice where used for experiments and what exposure was applied. Also, we now say that only ROFA exposed mice were used later and that this was the focus in this study (L221).

3. Accordingly, Figure 1 is misleading as it suggests all groups have been investigated throughout the experiment - please communicate the experimental setup clearly to the reader.

We modified Figure 1 together with comment #2 to communicate the experimental setup clearly to the reader.

---

## [Decision Letter · Decision Letter 2]

27 Aug 2020

Maternal supplementation with Bifidobacterium breve M-16V prevents their offspring from allergic airway inflammation accelerated by the prenatal exposure to an air pollutant aerosol

PONE-D-20-05759R2

Dear Dr. Ito,

We’re pleased to inform you that your manuscript has been judged scientifically suitable for publication and will be formally accepted for publication once it meets all outstanding technical requirements.

Kind regards,

Heinz Fehrenbach

Academic Editor

PLOS ONE

Additional Editor Comments (optional):

Reviewers' comments:

Reviewer's Responses to Questions

**Comments to the Author**

1. If the authors have adequately addressed your comments raised in a previous round of review and you feel that this manuscript is now acceptable for publication, you may indicate that here to bypass the “Comments to the Author” section, enter your conflict of interest statement in the “Confidential to Editor” section, and submit your "Accept" recommendation.

Reviewer #1: All comments have been addressed

Reviewer #2: (No Response)

2. Is the manuscript technically sound, and do the data support the conclusions?

Reviewer #1: Yes

Reviewer #2: Yes

3. Has the statistical analysis been performed appropriately and rigorously? 

Reviewer #1: N/A

Reviewer #2: Yes

4. Have the authors made all data underlying the findings in their manuscript fully available?

Reviewer #1: Yes

Reviewer #2: Yes

5. Is the manuscript presented in an intelligible fashion and written in standard English?

Reviewer #1: Yes

Reviewer #2: Yes

6. Review Comments to the Author

Reviewer #1: (No Response)

Reviewer #2: (No Response)

7. PLOS authors have the option to publish the peer review history of their article (what does this mean?). If published, this will include your full peer review and any attached files.

Reviewer #1: No

Reviewer #2: No

---

## [Editor Report · Acceptance letter]

2 Sep 2020

PONE-D-20-05759R2 

Maternal supplementation with *Bifidobacterium breve* M-16V prevents their offspring from allergic airway inflammation accelerated by the prenatal exposure to an air pollutant aerosol 

Dear Dr. Ito:

I'm pleased to inform you that your manuscript has been deemed suitable for publication in PLOS ONE. Congratulations! Your manuscript is now with our production department. 

Kind regards, 

on behalf of

Prof. Dr. Heinz Fehrenbach 

Academic Editor

PLOS ONE